# Purchasing of tobacco-related and e-cigarette-related products within prisons before and after implementation of smoke-free prison policy: analysis of prisoner spend data across Scotland, UK

Catherine Susan Best ![iD] , Ashley Brown ![iD] , Kate Hunt ![iD]

Institute for Social Marketing and Health, University of Stirling, Stirling, UK

**Correspondence to**
Dr Catherine Susan Best;
catherine.best2@stir.ac.uk

## ABSTRACT

**Objectives** To examine the effect of smoke-free prison policy implementation in November 2018 on purchasing patterns in the prison canteen (shop).

**Design** Interrupted time series.

**Setting** All 12 closed, publicly run prisons in Scotland, UK.

**Participants** People in custody (PiC) between August 2018 and end of March 2019 (n=11 944).

**Interventions** Implementation of smoke-free prisons policy.

**Outcome measures** Total spent on all products, nicotine-related products, and food and beverage products per week.

**Methods** Canteen data were provided for the period July 2018–September 2019 by the Scottish Prison Service. In a series of generalised linear mixed effects models, the amount spent before and after implementation of smoke-free prison policy was compared for all purchases in the time period, and for PiC identified as 'smokers' and 'non-smokers' from their pre-implementation tobacco purchasing patterns.

**Results** The amount spent on nicotine-related products significantly decreased from pre-implementation to post implementation (incident rate ratio (IRR) 0.40; 99% CI 0.33 to 0.51, p<0.001). However, total canteen spend did not change over the study period (IRR 0.92; 99% CI 0.84 to 1.00). Post implementation about 25% of previous 'smokers' total canteen spend was on nicotine-related products. The amount spent by previous 'smokers' on food and beverages increased from £8.67 (99% CI 8.23 to 9.13) pre-implementation to £10.24 in the post implementation period (99% CI 9.58 to 10.90).

**Conclusion** Although the amount of money previous 'smokers' in prison spent on nicotine-related products decreased after smoke-free policy, nicotine products still account for a large proportion of canteen spend among PiC in smoke-free prisons in Scotland. Results indicate that many PiC may continue to use nicotine in smoke-free prisons where e-cigarettes are permitted.

## INTRODUCTION

In the UK[1] and elsewhere,[2] people who are or have been in prison experience worse health than the general population. The prison

---

## Strengths and limitations of this study

► A strength of this study is that we had a complete record of purchase data from all publicly managed closed prisons in Scotland, UK.

► The main limitation of this study is that we were not able to link the purchase data to other information and therefore have to use purchase patterns (previous purchase of tobacco) as a proxy for smoking status.

► Another limitation is that there were local restrictions on how much individuals could spend on e-cigarettes.

---

population is disproportionately drawn from marginalised sections of society; the imprisonment process can intensify or alleviate risk factors for poor health. For example, smoking rates are higher in socioeconomically deprived groups[3] and even higher among people in custody (PiC) in prisons where smoking is still permitted; in Scotland 68% of PiC smoked in 2017,[4] resulting in high levels of secondhand smoke (SHS).[5 6] Due to concern about the effects of smoking on the health of PiC and occupational SHS exposure for staff, the sale and use of tobacco was prohibited in Scottish prisons from 30 November 2018.

While detained, basic food and toiletries are provided but PiC in Scotland can purchase additional food and beverage products, personal hygiene products and other items from lists for the 'canteen' (prison shop); items are then delivered to someone's room (cell). People who are sentenced have the opportunity to make purchases from the canteen once per week. Spending limits apply, which vary depending on whether someone is on remand or convicted and their privilege level in prison. PiC who participate

in employment or training in prison are paid a weekly wage (ranging from £5 to £21),[7] which can be spent in the canteen. Some PiC receive additional funds from family and friends. In anticipation of the smoke-free policy, tobacco was removed from the canteen in mid-November 2018. As part of a comprehensive implementation plan, selected e-cigarette products were added to the canteen list to assist PiC in the transition to a tobacco-free environment; some PiC were eligible for free 'starter' packs provided in November 2018.[8] Access to free combined behavioural and pharmacological treatments (eg, Nicotine Replacement Therapy) continued to be available in prisons in the lead up to and following implementation of the smoke-free policy.[9]

To date, implementation of smoke-free prisons has not included the provision of e-cigarettes in jurisdictions outside of the UK, although e-cigarettes have subsequently gone on sale in some US prisons which are smoke-free. Scotland was the first country to undertake a multi-phase, multi-methods research programme to evaluate the process and outcomes of smoke-free prison policy,[8] including immediate and longer term impacts on air quality (measured as levels of fine inhalable particles with diameters of 2.5 µm and smaller—$PM_{2.5}$)as a measure of SHS exposure[6 10] and the response of staff and PiC to smoke-free policy before[5 11] and after implementation.[12] The prison canteen is important for PiC, since, for example, it is a means through which they can purchase toiletries, which can enhance their sense of dignity in a potentially challenging environment, or phone cards, which enable them to keep in better touch with family and friends. The proportion of their limited available resources which are spent on nicotine-related products (tobacco-related and e-cigarette-related products, where allowed) in the canteen thus potentially has a large impact on the finances, resources and well-being in this vulnerable population. It is important to understand whether PiC spend less or more on e-cigarette products in the post-'ban' period than they did previously on tobacco, to inform understandings of the individual and wider public health consequences of smoke-free prison policies. Effects on broader patterns of prison canteen purchasing behaviour (eg, changes in spend on confectionery or sugar-sweetened beverages) may signal other unintended (positive or negative) impacts of smoke-free prison policy, with potential implications for levels of obesity within prisons. In Canada, Johnson and colleagues found that smoke-free prison policy was associated with weight gain among PiC[13] and that weight gain while in prison was more closely correlated with the type of food purchased from the canteen than with food provided to PiC.[14] Any increases in purchases of obesogenic food products following the removal of tobacco from prisons may be an important consideration for other jurisdictions preparing to implement smoke-free prison policy.

This study examines purchasing data from the prison canteen in all 12 closed publicly run prisons in Scotland before and after implementation of smoke-free prison policy in 2018. The objective was to determine whether amount spent on all products, nicotine-related products, and food and beverage products from the prison canteen changed as a result of the implementation of smoke-free prisons policy. We examine these effects for all purchases in the time period, and then separately for those PiC identified as 'smokers' and 'non-smokers' from their pre-implementation tobacco purchasing patterns.

## METHODS

The prison population in Scotland was around 8200 people in July 2019. 79% of the prison population had been sentenced and the remainder were untried or awaiting sentence.[15] Anonymised data on individuals' purchases were supplied by Scottish Prison Service (SPS). PiC within 12 of the 15 prisons in Scotland (ie, excluding the one publicly run 'open' and two privately run prisons) in the form of deidentified, password protected excel worksheets. The data set received from SPS comprised information for purchases made between 18 July 2018, 1 year after the SPS announced its intention to make all prisons smoke-free in November 2018, and 30 September 2019.

We coded every purchase into one of six product types (coding by CSB and initial coding reviewed by AB and KH): *tobacco-related* (including tobacco, cigarette papers, filters, lighters and cigarette rolling devices); *e-cigarette-related* (devices, e-liquids and chargers); *personal hygiene; food* (including sweets, snacks and beverages); *communications* (stamps, writing materials and phone cards); or *other*. There was very little ambiguity about the classification of purchases. Although the purchases had brief names in the data, these included the brand name and as there were distinct e-cigarette, tobacco, food and hygiene brands it was straightforward to classify.

We then calculated the total spend for each individual per week on all products combined and for each of these six product types. Finally, we created a combined measure of the total spend on any tobacco-related or e-cigarette-related products by each person, per week, referred to collectively hereafter as 'nicotine-related products'. These include manufactured cigarettes, rolling tobacco, papers, filters and other smoking paraphernalia and e-cigarette products such as devices, e-liquid refills and chargers. Nicotine replacement therapy was not included as although it was available through the canteen purchase levels were so low as to be negligible.

## Patient and public involvement

It was not possible to involve study participants in the design, or conduct, or reporting, or dissemination plans of our research. However, the qualitative research with PiC and staff conducted as part of this wider study, explored their views and experiences of vaping in prisons before and after the implementation of smoke-free prisons and is published elsewhere.[9 11]

## Analysis

We conducted a series of generalised linear mixed effects models to examine changes in the total spend in the canteen per week and on the amount spent on nicotine-related products over time. These models account for the dependencies among repeated measures for each individual over time by including an individual level random intercept. The effect of the implementation of smoke-free prison policy on 'canteen' purchasing behaviour, expressed as incident rate ratios (IRRs), was estimated using an interrupted time series approach. The time series was divided into three periods:

1. The period before the removal of tobacco products from the canteen list (ie, period 1: 29 July 2018–30 September 2018).
2. The period including the introduction of rechargeable e-cigarette products and removal of tobacco-related products (ie, period 2: 1 October 2018–31 January 2019).
3. The period after these changes were made (ie, period 3: 1 February 2019–31 March 2019).

The comparison of interest was between periods 1 and 3. The estimate of effects of period 2 versus period 1 was of less interest as there were several changes during this time related to the imminent implementation of smoke-free policy, including the addition of two brands of rechargeable e-cigarettes to the canteen list, the provision of free introductory e-cigarette packs to eligible, recorded smokers and a graded pricing structure for e-cigarette products during this period (see Brown et al[16] for more detail). Period 2 also includes the Christmas period, when canteen spending patterns are temporarily disrupted. For example, there may be two opportunities for purchase in one calendar week before Christmas and none the week after because Christmas falls on the day the canteen would usually be available. This distorts the median weekly spend for those weeks. Hence, the comparison of period 1 versus period 3 best captures the effect of the smoke-free policy and the introduction of rechargeable e-cigarette products in Scottish prisons.

The generalised mixed models included a factor variable for prison, a variable for week number (1–35), a factor variable for time period, and an interaction between the two. The distribution of the purchase data was highly skewed with lots of small purchases and relatively few high value purchases. The appropriate link function for the generalised linear model was determined by comparing the Akaike information criterion for mixed effects Poisson, negative binomial and linear models. For all analyses the negative binomial model (log link) was associated with the smallest information loss. Marginal effects for the implementation of smoke-free prison policy were computed for all models. Marginal effects are a way of demonstrating the impact of change in one variable (pre-implementation vs post implementation) based on a model containing features such as interactions and random effects.

We also sought to take account of smoking status and the considerable turnover in the prisoner population over time. To account for effects of turnover in prison population we conducted an analysis including only people who had made purchases in periods 1–3. In addition, in order to distinguish the impact on likely smokers and likely non-smokers among PiC prior to implementation, we identified those who had made any purchase of tobacco in period 1 as 'smokers'.

To understand overall canteen spend, and changes in purchasing over time, we therefore conducted analyses in three different samples:

1. Group 1: the full dataset of all purchases.
2. Group 2: people who had purchased tobacco before implementation ('smokers' in period 1) and made purchases through periods 2 and 3 (ie, those who were still imprisoned).
3. Group 3: people who had not purchased tobacco before implementation ('non-smokers in' period 1) who made purchases through periods 2 and 3.

As there were a very large number of observations included in the analyses, the statistical significance threshold was set at 0.01. Data management and cleaning were conducted in Python and data analysis in Stata V.15; graphs were generated in R (V.4.0.3).

## RESULTS

### Products

#### Tobacco-related products pre-implementation

The total range of tobacco-related products sold in the pre-implementation period is shown in table 1. For each product, the unit price, the total spent and the total number of units purchased within period 1 are also presented. No information is presented for the post implementation period, as no tobacco-related products were sold via the canteen during this period.

Of the 21 unique tobacco-related products available for purchase in the pre-implementation period, spending was heavily concentrated on a small number of products and one brand ('Brand 2') of rolling tobacco accounted for 48% (185175/387608.61) of all spending on tobacco-related products in period 1.

#### E-cigarette-related products

Table 2 shows the price per item, total amount spent and the quantity of items purchased for all e-cigarette-related products for sale via the prison canteen in the time periods of interest (period 1 and period 3). Sixteen e-cigarette-related products were available in the post implementation period (those not yet available in period 1 are shown as £0 spend in table 2). Again, most purchasing was concentrated on a small number of products. The most popular product was a tobacco-flavoured e-liquids at 18 mg strength ('Brand A'), which alone accounted for 53% (149795.20/280257.92) of all spending on e-cigarette-related products.

**Table 1** Tobacco products available for purchase from the canteen list prior to implementation of smoke-free policy

| Product | Price (£) | Pre-implementation period (period 1) | |
| --- | --- | --- | --- |
| | | Total spend 29 July 2018–30 September 2018 | Total quantity 29 July 2018–30 September 2018 |
| Pipe tobaccos | | | |
| Brand 1 pipe tobacco 12.5 g | 2.59 | 83 400.59 | 32 201 |
| Brand 9 pipe tobacco 50 g | 10.20 | 408.00 | 40 |
| Brand 7 pipe tobacco 25 g | 6.60 | 3042.60 | 461 |
| Rolling tobaccos | | | |
| Brand 2 rolling tobacco 30 g | 12.50 | 185 175.00 | 14 814 |
| Brand 5 rolling tobacco 30 g | 12.90 | 838.50 | 65 |
| Brand 6 rolling tobacco 30 g | 13.35 | 1174.80 | 88 |
| Brand 10 rolling tobacco 30 g | 10.00 | 32 450.00 | 3245 |
| Brand 8 rolling tobacco 30 g | 11.35 | 66 318.05 | 5843 |
| Factory made cigarettes/cigars | | | |
| Brand 3 king size 20 cigarettes | 9.60 | 2611.2 | 272 |
| Brand 8 20 cigarettes | 9.65 | 434.25 | 45 |
| Brand 4 cigar drum | 1.01 | 131.30 | 130 |
| Paraphernalia | | | |
| Disposable lighter | 0.10 | 2554.10 | 25 543 |
| RYO papers | 0.10–0.26 | 6725.94 | 54 937 |
| Filters | 0.40–0.58 | 2329.70 | 4069 |
| Rolling machine | 1.62 | 14.58 | 9 |
| Total | | 387 608.61 | |

**Table 2** Spend on e-cigarette-related products pre-implementation and post implementation of smoke-free policy

| Product | Pre-implementation (period 1) | | | Post implementation (period 3) | | |
| --- | --- | --- | --- | --- | --- | --- |
| | Price | Total spend 29 July 2018–30 September 2018 | Total quantity 29 July 2018–30 September 2018 | Price | Total spend 1 February 2019– 31 March 2019 | Total quantity 1 February 2019– 31 March 2019 |
| Devices | | | | | | |
| Brand A rechargeable device | 7.00 | 574.00 | 82 | 7.00 | 5565.00 | 795 |
| Brand B rechargeable device | 4.33 | 329.08 | 76 | 6.49 | 1090.32 | 168 |
| USB mains plug for rechargeable device | 4.00 | 216.00 | 54 | 4.00 | 1048 | 262 |
| Introductory e-cigarette pack | 0 | 0 | 0 | 7.10 | 1874.40 | 264 |
| Disposable e-cigarettes | 1.20 | 3554.20 | 1594 | 1.20 | 4465.80 | 3673 |
| Capsules | | | | | | |
| Brand A capsule—tobacco flavour 18 mg | 3.20 | 246.40 | 77 | 3.20 | 149 795.20 | 46 811 |
| Brand A capsule berrymint—12 mg | 3.20 | 320.00 | 100 | 3.20 | 44 422.40 | 13 882 |
| Brand A capsule menthol—12 mg | 3.20 | 99.20 | 31 | 3.20 | 24 902.40 | 7782 |
| Brand A capsule strawberry—12 mg | 3.20 | 358.40 | 112 | 3.20 | 13 129.60 | 4103 |
| Brand A capsule red cherry—12 mg | 0 | 0 | 0 | 3.20 | 10 665.60 | 3333 |
| Brand A capsule berry mint—6 mg | 0 | 0 | 0 | 3.20 | 451.20 | 141 |
| Brand B capsule strawberry 18 mg | 6.80 | 292.40 | 43 | 6.80 | 7718 | 1135 |
| Brand B capsule blackcurrant—18 mg | 6.80 | 244.80 | 36 | 6.80 | 15 130 | 2225 |
| Total | | 6234.84 | | | 280 257.92 | |

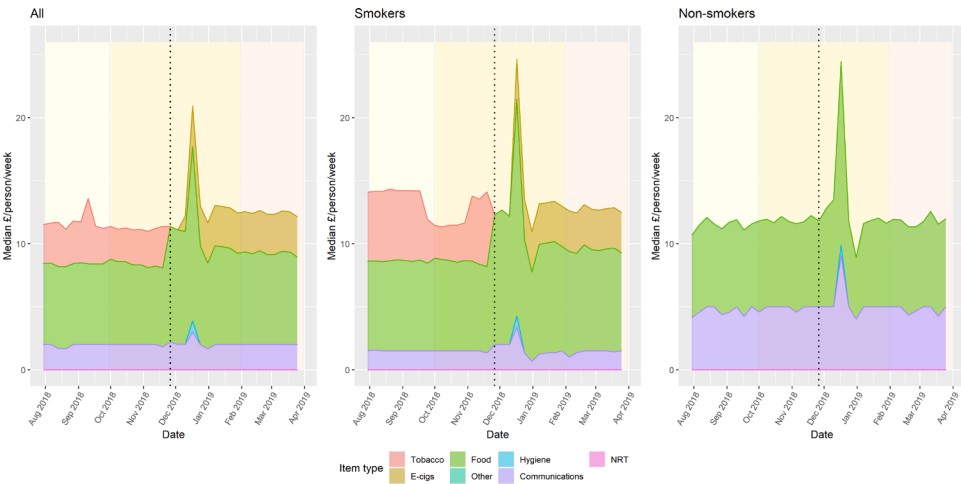

**Figure 1** Median weekly spend per person by product category.

## Median spend per person per week by product type

Figure 1 shows the median spend per person per week for the six purchasing categories (tobacco-related, e-cigarette-related, personal hygiene, food, communications and other). The first graph shows the composition of canteen spend for all people in the dataset (n=11 944), the second graph is for 'smokers' who were present in the dataset in all three periods (n=2541) and the third is for 'non-smokers' present across all the three time periods (n=342).

The dotted lines in figure 1 indicate the date of the introduction of smoke-free policy in Scottish prisons. The yellow shading indicates period 2 which included the one-off provision of free e-cigarette starter packs for eligible smokers; changes in purchasing patterns during the Christmas period are clearly seen in the spike in purchases in late December.

## Analysis of changes in canteen purchasing behaviour

We examined the relationship between the implementation of smoke-free prison policy and weekly canteen spend by individuals for three dependent variables, that is, individual canteen spend per week on: all products; nicotine-related products; and food and beverage products.

We examined these outcomes for: (i) the full sample, that is, all people who purchased anything from the canteen during the study period (n=11 994), (ii) (a) 'smokers' and (b) 'non-smokers' making purchases throughout the implementation period (periods 1, 2 and 3) defined as purchasing in at least 31 of 35 weeks (n=2541). Table 3 shows results for three groups.

The results in table 3 indicate that there was no notable change in the total spend per person per week on canteen products (IRR 0.92; 99% CI 0.84 to 1.00) following implementation of smoke-free prison policy: the marginal

**Table 3** Results from mixed effects negative binomial regression examining changes in canteen spend pre-implementation vs post implementation of smoke-free prison policy

| | Variable | Total spend (£)<br>IRR (99% CI) | Nicotine-related products (£)<br>IRR (99% CI) | Food items (£)<br>IRR (99% CI) |
|---|---|---|---|---|
| Full sample n=11 944 (group 1) | Prechange vs postchange | 0.92 (0.84 to 1.00) | 0.40 (0.33 to 0.51)† | 1.12 (0.99 to 1.27) |
| | Change in time trend | 1.00 (1.00 to 1.00) | 1.02 (1.01 to 1.03)† | 1.00 (1.00 to 1.01) |
| | Pretime trend | 1.00 (1.00 to 1.00) | 0.99 (0.99 to 1.00) | 1.00 (0.99 to 1.00) |
| 'Smokers' resident within a prison for more than 31 weeks (periods 1–3) n=2541 (group 2) | Prechange vs postchange | 0.91 (0.81 to 1.03) | 0.35 (0.25 to 0.50)† | 1.24 (1.08 to 1.43)* |
| | Change in time trend | 1.00 (1.00 to 1.01) | 1.02 (1.01 to 1.04)† | 1.00 (0.99 to 1.00) |
| | Pretime trend | 1.00 (1.00 to 1.00) | 0.99 (0.99 to 1.00) | 1.00 (1.00 to 1.00) |
| 'Non-smokers' resident within a prison for more than 31 weeks n=342 (group 3) | Prechange vs postchange | 0.87 (0.66 to 1.16) | 0.08 (0.00 to 1.62) | 0.80 (0.540 to 1.19) |
| | Change in time trend | 1.01 (0.99 to 1.02) | 1.18 (1.03 to 1.36) | 1.01 (0.99 to 1.02) |
| | Pretime trend | 1.00 (0.99 to 1.01) | 1.03 (0.93 to 1.14) | 1.00 (0.99 to 1.01) |

*p<0.01
†p<0.001
IRR, incident rate ratio.

predicted total canteen spend per person per week in the full sample in the pre-implementation period was £20.46 (99% CI 19.98 to 20.94) and the marginal predicted spend in the post implementation period was £19.43 (99% CI 18.88 to 19.98).

The implementation of smoke-free policy was associated with a decrease in the amount spent per week on nicotine-related products or NVPs (p<0.001). This was evident in both the full sample (group 1) and those who were identified as 'smokers' in the pre-implementation period and were resident in prison for the full period under study (periods 1–3) (group 2). The marginal predicted spend on nicotine-related products in the pre-implementation period in these 'smokers' was £9.23 (99% CI 8.35 to 10.12) and in the post implementation period was £5.21 (99% CI 4.66 to 5.77). In the post implementation period (period 3) the amount individuals spent on nicotine-related products showed a slight statistically significant week by week increase.

There was no change in the amount spent on nicotine-related products by people designated as 'non-smokers'. That is, there is no indication people who had *not* purchased tobacco during period 1 had started purchasing e-cigarette-related products in the post implementation period.

In contrast, the amount spent on food and beverage products increased in the post implementation period. This effect was only significant in group 2 ('smokers' who were resident in the prison over for the full period under study (periods 1–3)). The marginal predicted spend on food in the pre-implementation period by these 'smokers' increased from £8.67 (99% CI 8.23 to 9.13) to £10.24 in the post implementation period (99% CI 9.58 to 10.90).

## DISCUSSION

This study has shown that on average, following the introduction of smoke-free prison policy, PiC in Scotland spend less per week on canteen purchases for e-cigarette-related products than they previously did on tobacco-related products. The IRR for the change is 0.35 (99% CI 0.25 to 0.50). This decline is also evident in total spend on tobacco-related and e-cigarette-related products at £393 843.45 and £280 257.92 in the pre-implementation and post implementation periods, respectively.

Nevertheless, the amount spent on nicotine-related products still represents a large proportion of the total amount spent in the canteen by PiC, with most previous 'smokers' spending around 25% of their total spend per week in the canteen on e-cigarette-related products following implementation of smoke-free policy. This concurs with findings from qualitative interviews with PiC in Scotland. On the positive side, interviewees who were spending less on e-cigarettes post implementation of smoke-free policy than they had previously spent on tobacco described benefits such as being able to buy healthier food from the canteen or other valued items, or being able to save money. However, there were beliefs

that e-cigarettes were not affordable relative to the income for PiC with the fewest resources. Additionally, some complaints were made about the high upfront costs of vaping in prison and that the e-cigarette-related products available on the canteen list at that time represented poor value for money, for example, because they were perceived as less satisfying than conventional cigarettes or as compared with costs of e-cigarette products in the community.[9]

The amount spent on food and beverages by previous 'smokers' increased in the post implementation period relative to pre-implementation. The IRR for the change was 1.24 (99% CI 1.08 to 1.43). This possibly indicates snacking was used as a displacement activity following the removal of tobacco[17 18] and/or that people were able to diversify their discretionary food purchases due to saving money on nicotine-related products. Further research is required to examine differences in the breakdown of the food purchasing activity within food groups pre-implementation and post implementation, and whether this impacts on body mass index and other health-related indicators.

We found that although a range of tobacco products were available pre-implementation, purchasing was concentrated in a very small number of products, dominated by rolling tobacco. Post implementation, nicotine-related purchasing was also concentrated on a narrow range of products, for example, the most popular being the cheapest and highest strength e-liquid (18 mg/mL). This possibly indicates that previous smokers were trying to maximise nicotine intake while minimising spending.

## Limitations

It was not possible to link the canteen data to other sources of information about individuals (eg, age, gender, custody status, smoking status) and we were therefore restricted to making assumption about whether individuals were smokers or not based on their purchasing patterns pre-implementation. We were also not able to link to information on smoking cessation support or the provision of prescribed NRT within the prisons' smoking cessation services.

There are local restrictions on the total amount of money PiC were permitted to spend on e-cigarettes post implementation, set at the discretion of prison Governors. Information on the restrictions at individual prisons is not in the public domain. However, we understand that the guidance to Governors on limiting e-cigarette purchasing suggested maximums far in excess of the median spend on e-cigarette products observed here.

The strengths of this study are that we have complete data from the canteen spend in closed publicly operated prisons in Scotland for a time period spanning the introduction of smoke-free prison policy. This is unique data, and we are not aware of any other studies within the UK or internationally that have been able to study the spending choices of PiC or to evaluate the effects of smoke-free prison policy on canteen spending. This research will be

important for jurisdictions considering implementing smoke-free prisons in the future.

Our conclusions are limited to the amount of money that PiC spent on tobacco and e-cigarettes pre-implementation and post implementation. This study did not aim to estimate how much nicotine was consumed and whether this differed from pre-implementation to post implementation. This would be challenging to estimate from canteen purchases given that the amount of nicotine that becomes bio-available varies enormously across products and methods of use. Effect on nicotine consumption would be an important topic for future research as it would have implications for smoking and vaping behaviour postrelease.

## Conclusion

Although smoke-free policy has decreased the amount of money previous 'smokers' in prison spend on nicotine-related products after implementation of smoke-free policy, nicotine-related products still account for a large proportion of discretionary purchases by PiC. Ongoing monitoring is required to understand the longer-term benefits and risks of selling e-cigarettes in prisons, including how vaping affects the personal finances and health of PiC and influences smoking behaviour postrelease.

**Acknowledgements** We are grateful to staff at the Scottish Prison Service who assisted with the study and facilitated access to data. We are also grateful to the wider Tobacco In Prisons Study research team.

**Contributors** AB and KH: conception of the work, acquisition of data, communication and collaboration with Scottish Prison Service, interpretation of data, and revision of manuscript for important intellectual content. CSB: management and analysis of the data, interpretation of data, creating first draft of manuscript and revision of the manuscript. CSB is the author who is guarantor.

**Funding** The study was supported by Cancer Research UK (C45874/A27016).

**Competing interests** None declared.

**Patient and public involvement** Patients and/or the public were not involved in the design, or conduct, or reporting, or dissemination plans of this research.

**Patient consent for publication** Not applicable.

**Ethics approval** This study involves human participants and this study was reviewed and approved by the General University Ethics Panel of the University of Stirling ref GUEP497(A).The study is secondary analysis of administrative data. It would not be possible to link this data back to an individual person.

**Provenance and peer review** Not commissioned; externally peer reviewed.

**Data availability statement** Data may be obtained from a third party and are not publicly available. The data sets were made available by the Scottish Prisons Service for the purposes of this analysis.

**ORCID iDs**
Catherine Susan Best http://orcid.org/0000-0002-3652-2498
Ashley Brown http://orcid.org/0000-0002-2307-5916
Kate Hunt http://orcid.org/0000-0002-5873-3632

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
