## [Reviewer comments · BMJ Open]

ARTICLE DETAILS

TITLE (PROVISIONAL)	Purchasing of tobacco-related and e-cigarette-related products within prisons before and after implementation of smoke-free prison policy: analysis of prisoner spend data across Scotland.
AUTHORS	Best, Catherine; Brown, Ashley; Hunt, Kate

VERSION 1 – REVIEW

REVIEWER	Walsh, Hannah King's College London, Florence Nightingale Faculty of Nursing, Midwifery and Palliative Care
REVIEW RETURNED	10-Dec-2021

GENERAL COMMENTS	Thank you for the opportunity to review this interesting and important paper on the topic of tobacco and nicotine product spend in Scottish prisons, using time-series analysis to examine various changes in spend pre and post smokefree legislation. I find the paper to be well-written and clear, the statistical analysis and choices made around time periods appropriate, and the results are clearly presented. I have only few suggestions to improve clarity and expand the reader's understanding of this specific context. In relation to the standard questions above: Question 1,6: The paper would benefit from an explicit statement regarding the study objectives, which expands on the stated aim. The aim and rationale for exploring this topic is provided and the objectives may be inferred from this, but it would add clarity to specify the objectives. Qu 9: see points above Additional suggestions: Abstract results section: to amplify the key finding of interest, you could switch the order of the first two sentences. Introduction: it may be helpful and certainly of interest to the reader to expand a little on the process of purchasing items at the canteen; eg does each PiC have the same amount of money to spend each week, or does this differ across individuals, and what might be the variation in spend; is there a maximum permitted spend? Is the canteen open daily or weekly only? When the PiC accesses the canteen, do they pre-select items from a list, or do they view items on display as in a regular shop? This would help the reader understand the specific purchasing context of custody. It would help to understand a little more about NRT availability, e.g. what smoking cessation treatment is provided in prisons, does
--

	this include NRT at all, or is the only possible access to NRT through canteen purchases? What is the influence of Christmas - i.e. do PiC receive more money at this time? Do some of them have more visitors at this time? Does the routine change which may explain higher purchases of items? On page 4, l48 expand 'PM2.5' at first use. Ethical approval: the authors may consider giving a very brief explanation as to why it wasn't possible to include study participants - it is completely understandable and unsurprising - but a brief explanation may enhance the reader's understanding of the challenges of carrying out research in prisons, and it could therefore demonstrate the added value of a study such as this one. Discussion: p13 l18: suggest rewording this paragraph to avoid the frequent use of limit/ limitation, and to clarify that it is the amount given as a limit and not the guidance which was in excess of median spend. l28: it would be helpful and add value if the authors could expand a little on why it would be (potentially) useful to understand how much nicotine was consumed, does this constitute a limitation, or is this an avenue for further research? The study has significant strengths, not least the unique data set; the final paragraph on p13 could showcase these a little more. Graphs: If the TSA graph could be presented at slightly larger scale it would enhance it; once enlarged it is very well presented.
--	---

REVIEWER	Shoesmith, Emily University of York
REVIEW RETURNED	13-Dec-2021

GENERAL COMMENTS	The authors investigate the purchasing of tobacco-related and e-cigarette-related products within prisons before and after implementation of smoke-free prison policy. This is a very novel study with very interesting findings. The authors present a well-structured and clear introduction, providing a strong rationale for the aim of the study. The discussion is well grounded within the current findings and appropriately explores the findings in the wider context. I had a number of thoughts about limitations while reading the manuscript, but these are clearly addressed in the limitation section. While the authors could not link the data to other sources of information (which does restrict the results and subsequent interpretation), I believe the reported findings are still a beneficial contribution to the field and provide a strong foundation for further research to be built on. I just have one minor comment for the methods: I would suggest clarifying exactly who coded each purchase and how, e.g., was it all authors independently? Did one author independently code and this was reviewed by other authors? This is a timely and beneficial contribution to the field and I look forward to seeing it published.
---

VERSION 1 – AUTHOR RESPONSE

Reviewer: 1

Thank you for the opportunity to review this interesting and important paper on the topic of tobacco and nicotine product spend in Scottish prisons, using time-series analysis to examine various changes in spend pre and post smokefree legislation. I find the paper to be well-written and clear, the statistical analysis and choices made around time periods appropriate, and the results are clearly presented.

Thank you for your comments.

The paper would benefit from an explicit statement regarding the study objectives, which expands on the stated aim. The aim and rationale for exploring this topic is provided and the objectives may be inferred from this, but it would add clarity to specify the objectives.

Study objectives have now been added at the top of page 6 in the marked-up version of the manuscript.

Additional suggestions:

Abstract results section: to amplify the key finding of interest, you could switch the order of the first two sentences.

Thank you- we have changed the order of these sentences.

Introduction: it may be helpful and certainly of interest to the reader to expand a little on the process of purchasing items at the canteen; eg does each PiC have the same amount of money to spend each week, or does this differ across individuals, and what might be the variation in spend; is there a maximum permitted spend? Is the canteen open daily or weekly only?

We have added additional text in the Introduction on page 4 paragraph 2 to explain more about the process of purchase from the prison canteen for people in custody. Although we know that spending limits apply to people in custody depending on whether they are on remand or convicted the figures for these limits are not in the public domain. People in custody who have been convicted can purchase from the canteen once per week.

When the PiC accesses the canteen, do they pre-select items from a list, or do they view items on display as in a regular shop? This would help the reader understand the specific purchasing context of custody.

We have added text to the Introduction on page 4 paragraph 2 to give more information on how purchases are made. People in custody are given a list of items they can choose from and then the items are delivered to their room (cell).

It would help to understand a little more about NRT availability, e.g. what smoking cessation treatment is provided in prisons, does this include NRT at all, or is the only possible access to NRT through canteen purchases?

NRT is available through the prison smoking cessation services. We have added an extra sentence to indicate this at the end of paragraph 2 page 4. We have also added a reference with a link to a document where readers can see the full details of the cessation services offered to people in custody during the transition to smoke-free prisons.

What is the influence of Christmas - i.e. do PiC receive more money at this time? Do some of them have more visitors at this time? Does the routine change which may explain higher purchases of items?

One of the main causes of the peak in median spend is that many people in custody receive two opportunities to purchase from the canteen in one calendar week just before Christmas, in order to make up for no canteen on Christmas Day. That is, the canteen purchase might be a day early and then no purchase in the following week because of the way Christmas falls. We have added a sentence on this on page 8 paragraph 1. There will be other factors as well, such as, as the reviewer suggests, receiving money from family. If it had just been that the canteen was a few days early we could have corrected for that alone but there are likely to be many related factors that make Christmas unrepresentative of normal purchasing patterns.

On page 4, l48 expand 'PM2.5' at first use.

We have changed PM2.5 to 'air quality (measured as levels of fine inhalable particles with diameters of 2.5 micrometres and smaller - PM2.5)'.

Ethical approval: the authors may consider giving a very brief explanation as to why it wasn't possible to include study participants - it is completely understandable and unsurprising - but a brief explanation may enhance the reader's understanding of the challenges of carrying out research in prisons, and it could therefore demonstrate the added value of a study such as this one.

We have added to the PPI statement that we have carried out qualitative research with PiC and prison staff as part of the wider study, and that findings are reported elsewhere e.g. Brown A, O'Donnell R, Eadie D, Ford A, Mitchell D, Hackett A, Sweeting H, Bauld L, Hunt K. E-cigarette use in prisons with recently established smokefree policies: a qualitative interview study with people in custody in Scotland. *Nicotine & Tobacco Research*, Vol 23:6. <https://doi.org/10.1093/ntr/ntaa271>, Brown A/O'Donnell R (joint first authors), Eadie D, Purves R, Sweeting H, Ford A, Bauld L, Hunt K. 2020. Initial Views and Experiences of Vaping in Prisons: A Qualitative Study With People in Custody Preparing for the Imminent Implementation of Scotland's Prison Smokefree Policy. *Nicotine & Tobacco Research*, ntaa088.

Discussion:

p13 l18: suggest rewording this paragraph to avoid the frequent use of limit/ limitation, and to clarify that it is the amount given as a limit and not the guidance which was in excess of median spend.

Thank you we have revised this paragraph (page 14 lines 3-8) to avoid repetition of limit/ limitation.

l28: it would be helpful and add value if the authors could expand a little on why it would be (potentially) useful to understand how much nicotine was consumed, does this constitute a limitation, or is this an avenue for further research?

Thank you for this suggestion we have moved this paragraph to the bottom of page 14 and reworded it as a suggestion for future research rather than a limitation. The reason this was added as a limitation is that when this research has been presented to colleagues in public health and at research conferences, we were asked why we had not looked at total nicotine availability as this would have implications for dependence. However, in this data it is very difficult to assess the amount of nicotine that is bioavailable as there is large variation between products and by vaping/ smoking patterns of use. For this reason, we have been very careful to state the aims and objectives of this work as being to assess effects on purchase patterns rather than use and so in that context we agree with the reviewer that it is not a limitation.

The study has significant strengths, not least the unique data set; the final paragraph on p13 could showcase these a little more.

Thank you we have added another sentence on the importance of this research for other jurisdictions considering implementing smoke free prisons.

Reviewer: 2

The authors present a well-structured and clear introduction, providing a strong rationale for the aim of the study. The discussion is well grounded within the current findings and appropriately explores the findings in the wider context. I had a number of thoughts about limitations while reading the manuscript, but these are clearly addressed in the limitation section. While the authors could not link the data to other sources of information (which does restrict the results and subsequent interpretation), I believe the reported findings are still a beneficial contribution to the field and provide a strong foundation for further research to be built on.

Thank you for your comments.

I just have one minor comment for the methods: I would suggest clarifying exactly who coded each purchase and how, e.g., was it all authors independently? Did one author independently code and this was reviewed by other authors?

The initial coding was by CB and reviewed in early stages by AB and KH. There was very little ambiguity about the classification of purchases. Although the purchases had brief product names in the data-these included the brand name and as there were distinct e-cigarette and tobacco brands it was straightforward to classify. We have added this information on page 6 paragraph 2 of the methods. For example, 'FOSTERS PIPE 12.5g' as tobacco and 'LOGIC PRO CAP BERRYMINT - 12MG' as e-cig capsules. Food was also straight forward to distinguish from hygiene products e.g. MCVITIES HOB NOBS versus ALBERTO SHAMPOO RASPBERRY.

VERSION 2 – REVIEW

REVIEWER	Walsh, Hannah King's College London, Florence Nightingale Faculty of Nursing, Midwifery and Palliative Care
REVIEW RETURNED	27-Jan-2022
GENERAL COMMENTS	Thank you for the opportunity to review this revised manuscript. The authors have addressed all points raised in the review process appropriately, and I consider this paper suitable for publication.